# Align Your Prompts: Test-Time Prompting with Distribution Alignment for Zero-Shot Generalization

**Jameel Hassan**[1*]   **Hanan Gani**[1*]   **Noor Hussein**[1]   **Muhammad Uzair Khattak**[1]
**Muzammal Naseer**[1]   **Fahad Shahbaz Khan**[1,2]   **Salman Khan**[1,3]
[1]Mohamed Bin Zayed University of AI    [2]Linköping University    [3]Australian National University
{jameel.hassan, hanan.ghani, noor.hussein, uzair.khattak
muzammal.naseer, fahad.khan, salman.khan} @mbzuai.ac.ae

## Abstract

The promising zero-shot generalization of vision-language models such as CLIP has led to their adoption using prompt learning for numerous downstream tasks. Previous works have shown test-time prompt tuning using entropy minimization to adapt text prompts for unseen domains. While effective, this overlooks the key cause for performance degradation to unseen domains – distribution shift. In this work, we explicitly handle this problem by aligning the out-of-distribution (OOD) test sample statistics to those of the source data using prompt tuning. We use a single test sample to adapt multi-modal prompts at test time by minimizing the feature distribution shift to bridge the gap in the test domain. Evaluating against the domain generalization benchmark, our method improves zero-shot top-1 accuracy beyond existing prompt-learning techniques, with a $3.08\%$ improvement over the baseline MaPLe. In cross-dataset generalization with unseen categories across 10 datasets, our method improves consistently across all datasets compared to the existing state-of-the-art. Our source code and models are available at https://jameelhassan.github.io/promptalign/.

## 1   Introduction

Deep neural networks (DNNs) have outperformed humans in numerous visual recognition tasks [12, 8]. However, their impressive performance generally holds in the case when test data originate from the same distribution as the training data. In most real-world applications, the train and test data distributions can significantly differ due to factors such as natural variations or changes in sensing equipment. The sensitivity of models to such unseen distributional shifts during inference results in their performance degradation [14, 31, 29]. A plethora of previous works [36, 39, 37] have explored test-time adaptation as a mechanism to overcome the crucial problem of distribution shift in test data. However, test-time adaptation has been minimally explored for the increasingly popular foundation models, which are large DNNs trained on massive vision-language datasets [30, 17, 44, 40, 41].

Foundation models emerging at the intersection of various modalities such as vision, language, and audio, are proving to be effective in numerous downstream applications. Among these models, Vision-Language (V-L) model CLIP (Contrastive Language-Image Pretraining) [30] has been pre-trained on large-scale image-text pairs from the web, and can generalize well to zero-shot recognition tasks. The CLIP model has parallel vision and language encoding branches and the similarities between the embeddings obtained from both the branches are used to classify an input image. At inference, a handcrafted prompt such as 'a photo of a <CLS>' is used as a query for the text encoder. However, adapting CLIP efficiently to specific downstream tasks is still a challenging problem. Naively fine-tuning such models poses the risk of losing their inherent generalization abilities [21]. Instead, recent

---

*Equal contribution

37th Conference on Neural Information Processing Systems (NeurIPS 2023).

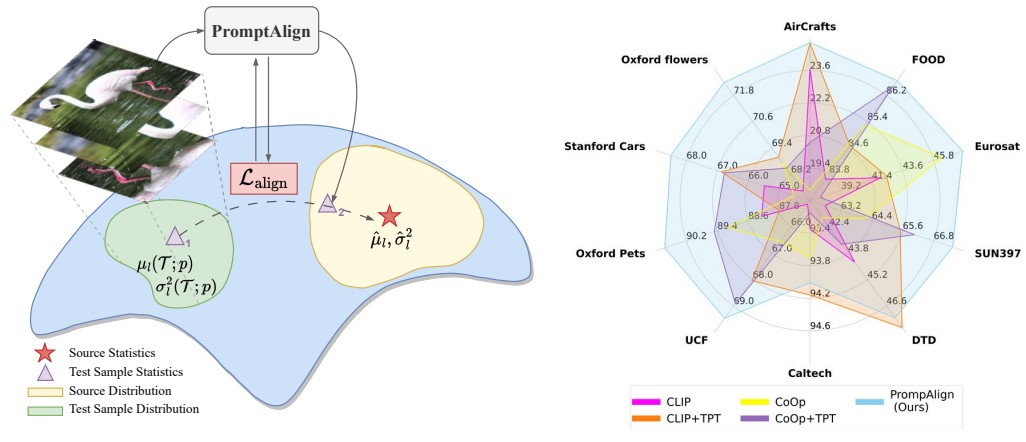

(a) Proposed PromptAlign method    (b) Performance comparison in cross datasets evaluation

Figure 1: (a) PromptAlign matches the distribution statistics $\boldsymbol{\mu}_l(\mathcal{T}; \boldsymbol{p})$, $\boldsymbol{\sigma}_l^2(\mathcal{T}; \boldsymbol{p})$, obtained from multiple augmented views of a single test sample, with the source data distribution statistics $\hat{\boldsymbol{\mu}}_l$, $\hat{\boldsymbol{\sigma}}_l^2$. This effectively brings the test sample closer to the distribution of the source data, where the domain shift is denoted by $\blacktriangle_1 \rightarrow \blacktriangle_2$. $\mathcal{T}$ denotes the distribution of the test sample, $\boldsymbol{p}$ represents the prompts that are updated and $l$ refers to the vision-backbone layers. (b) Owing to the distribution matching via prompts, PromptAlign surpasses the existing state-of-the-art prompt learning approaches on 8 out of 10 datasets in cross-dataset generalization benchmarks.

methods have shown the promise of prompt learning on training data [48, 47, 18, 19, 42] as opposed to handcrafted prompts, allowing the model to better adapt to the training data distribution.

Existing prompt learning approaches are deployed at the training phase to learn representative prompts based on the training data for the downstream task. This conventional approach does not explicitly handle the distribution shift in the test set. Leveraging the capability of prompt learning, a recent approach TPT [34] performs test-time adaptation by tuning the text prompts on the fly to adapt the model to the test sample. For image classification, the model updates the prompts by minimizing the entropy of the top confident samples which are obtained using different augmented views. However, TPT does not explicitly align the pre-trained CLIP to become aware of the test sample distribution.

For the effective test-time adaptation of V-L foundation models, it is crucial to bridge the distribution gap between the pre-training dataset and the downstream evaluation set for high zero-shot generalization. To this end, we propose PromptAlign, a test-time token distribution alignment strategy using prompt learning. The TPT setting, which tunes the prompts on the text encoder branch alone poses an architectural limitation in performing distribution alignment of tokens. Further, at test time, there is no knowledge transfer between the text and vision branches in the intermediate stages. Since the text encoder features will be static given a dataset (as the input is the same class labels), distribution alignment of tokens can only be performed on the vision branch. Given these constraints and to further extend the strength of prompts for test time adaptation, we propose to employ a multi-modal prompt learning model (MaPLe) [18] for distribution alignment. PromptAlign explicitly aligns the mean and variances of the image token embeddings of a proxy source dataset, computed offline, with the image token embeddings of the test sample. We extend TPT with the token alignment strategy, which enforces to bridge the distribution shift in the test data (Fig. 1 a). For each input test sample, we obtain randomly augmented views that are fed into the model to obtain the token embedding statistics. Without any noticeable compute overhead, we update the prompts on both the text and vision branches of CLIP to minimize jointly the feature distribution shift and entropy of predictions.

We extensively demonstrate the effectiveness of PromptAlign by evaluating the zero-shot generalization on two representative benchmarks: domain generalization and cross-dataset generalization. In the domain generalization setting, our method improves the baseline model by $3.08\%$ on average across the four datasets and has the highest average Top-1 accuracy compared to existing state-of-the-art methods. In the cross-dataset setting, our method achieves an absolute average improvement of $1.82\%$ over the existing state-of-the-art method which uses test-time prompt tuning, while attaining the best Top-1 accuracy in 8 out of the 10 datasets. Our contributions can be summarized as follows:

- Given only a single test sample, we introduce a distribution alignment strategy for V-L models to improve test-time adaptation. The distribution-aware pre-trained CLIP effectively narrows the distribution gap on the test domains. To the best of our knowledge, this is the first study to explore the potential of distribution alignment in V-L models at test time.

- The proposed strategy formulates a distribution alignment loss that utilizes offline computed source data statistics to encourage the test sample token distributions to be aligned with the source data token distributions. We harmonically combine the benefits of token distribution alignment and entropy minimization using a multi-modal prompt learning approach.

- Since CLIP-pre-training data is not publicly released, we study the statistics of ImageNet as a possible candidate for the source distribution, and our empirical results show that ImageNet is an effective proxy source dataset for large-scale V-L models such as CLIP.

- We validate our method PromptAlign through extensive experiments in domain generalization and cross-dataset benchmarks. PromptAlign improves the generalization of CLIP at test time beyond existing prompt-tuning methods, achieving state-of-the-art results.

## 2 Related Work

**Prompting for Vision-Language models.** Vision Language (V-L) foundation models [30, 17, 44, 40, 41] have emerged with the convergence of image and text modalities. Having been pre-trained on massive image-text pairs from the web in a contrastive self-supervised manner, these models like CLIP [30] and ALIGN [17] have shown strong generalizability towards downstream zero-shot recognition tasks. However, adapting them efficiently to specific downstream tasks with limited data is still a challenging problem. Prompting in CLIP-like models has been explored in the form of a text query that is usually given to the text encoder to instruct it to perform a specific task. Recent methods propose to learn these prompts by treating them as continuous learnable vectors and training them in an end-to-end manner while keeping the model parameters frozen. CoOp [48] proposes to fine-tune CLIP by learning a set of prompts in the text encoder. CoCoOp [47] highlights the inferior generalization capability of CoOp and conditions the text prompt tokens on image embeddings on the fly. MaPLe [18] proposes to jointly learn deep prompts at both vision and text encoders of CLIP. The vision prompts are further conditioned on text prompts via a V-L coupling function. While all these approaches promote the effective transfer of CLIP, they require training data to learn the prompts which restricts such adaptation on novel datasets at test time. A recent method TPT [34], attempts to learn prompts solely at test time with the objective of enforcing consistency regularization between multiple views of a test sample by minimizing their averaged entropy. However, this approach struggles to explicitly address the distribution misalignment between the pre-training data of CLIP and the downstream test data. Our method builds on multi-modal prompting variant [18] and explicitly enforces to match the distribution via a distribution alignment technique using a proxy dataset that serves as a replacement for unavailable pre-training data of CLIP. To the best of our knowledge, ours is the first work to explicitly align the distributions learned by V-L foundational models with test time data distribution.

**Test-Time Adaptation (TTA).** TTA [36, 37, 23] aims to bridge the distribution gap between the train and test data distributions at test time. These methods can broadly be categorized into two streams. The first stream of approaches [11, 36] typically utilizes a self-supervised proxy task such as predicting image rotations and thus alters the training phase. For example, Test-Time Training (TTT) [36] jointly trains the model for rotation prediction and image classification during training. TTT++ [23] adopts a self-supervised contrastive learning approach as an auxiliary task during training and test-time feature alignment module to adapt the model. By incorporating additional auxiliary tasks, these techniques aim to mitigate over-fitting and improve the generalization performance of the model. The second stream of approaches solely adopts the pre-trained model without altering the training phase. These methods improve model generalization at test time by imposing self-consistency regularization constraints or aligning statistics of training and test samples in a model. Approaches with self-consistency regularization constraints typically utilize entropy minimization within samples of a batch or multiple views of a single test sample. TENT [37] proposes entropy minimization of batch-wise prediction probability distributions. The need to have a batch of samples by generating multiple views via augmentations at test time is eliminated in [45]. Other methods, such as NORM [32] and DUA [25] adapt the model's normalization layers by matching the batch normalization statistics computed during training with the test set distribution. Both [22] and ActMAD

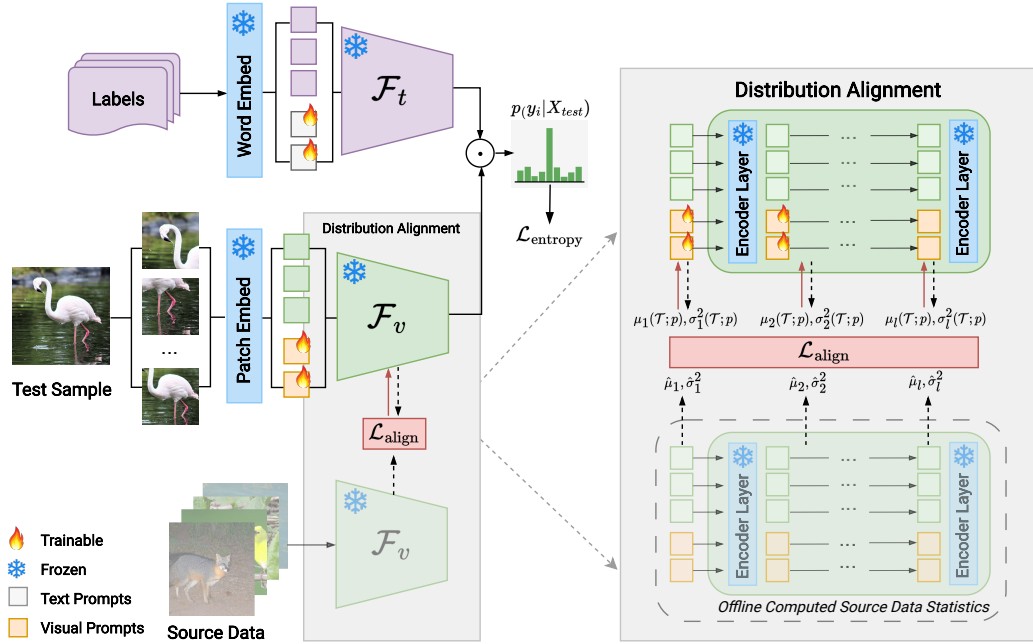

Figure 2: **Overview of our proposed PromptAlign method for zero-shot image classification.** At test time, a single test sample along with its augmented views is passed through the CLIP image encoder, and the text labels are passed to the CLIP text encoder. The token distribution statistics – mean and variance – of the test sample are aligned with the offline computed source data statistics using a distribution alignment loss. The resulting alignment loss from the distribution shift is combined with the entropy loss to update the multi-modal prompts.

[26] updates all model parameters using a batch of samples in a continuous manner, with a distribution matching loss between the source statistics and the batch statistics of activations across the network. However, it is unclear how these statistics alignment techniques can be adopted for foundational V-L models in a lightweight manner for a single sample. In this paper, we propose a multi-modal test time prompting approach without altering the pre-training phase of V-L models along with a strategy to obtain training distribution statistics. This effectively paves the way for distribution alignment together with entropy minimization for improved test-time optimization.

**Test-time prompt tuning.** As mentioned earlier, a plethora of works has been introduced on learning prompts to adapt V-L models using the downstream task training data. However, the learning of prompts at test time remains largely unexplored in the literature. Very recently, TPT [34] proposed a test time prompting approach by learning prompts at the text side with an entropy minimization objective. While effective, TPT struggles with addressing the explicit alignment of the pre-training data distribution of CLIP with the test data distribution. Further, such an alignment is not trivial as access to the pre-training data of CLIP is not available. Our work addresses these limitations by extending TPT and introducing a distribution alignment objective for CLIP. This objective leverages token distribution statistics from proxy source data and a *single* test sample using multi-modal prompt learning, and explicitly aligns the train and test sample distributions to mitigate the domain shift.

# 3 Methodology

We provide an overview of CLIP [30] and prompt learning and their application for zero-shot generalization to downstream tasks in Section 3.1. We then discuss test-time prompt tuning and briefly review previous approaches. Finally, we introduce PromptAlign in Section 3.2 and provide a detailed explanation of how we apply it to achieve improved zero-shot image classification.

### 3.1 Preliminaries

**Contrastive Language-Image Pre-training (CLIP).** CLIP consists of two parallel encoders, one for mapping the text input into a feature vector and the other for mapping the visual input. We denote the CLIP image and text encoders by $\mathcal{F}_v$ and $\mathcal{F}_t$ respectively, and their pre-trained parameters are represented by $\theta_{\text{CLIP}} = \{\theta_v, \theta_t\}$ respectively. The input image $X$ is divided into $M$ patches, which are projected to produce patch tokens, and a class token CLS is prepended to it, resulting in $\tilde{X}_0 = \{\text{CLS}, e_1, e_2, \ldots, e_M\}$ where $e_i$ is the embedding of the $i^{th}$ patch. The image encoder $\mathcal{F}_v$ encodes the input patches via transformer blocks to produce a latent visual feature representation $\tilde{f}_v = \mathcal{F}_v(\tilde{X}_0, \theta_v)$. The corresponding class label $y$ is embedded within a text template, such as 'a photo of a <CLS>', which is formulated as $\tilde{Y}_0 = \{\text{SOS}, t_1, t_2, \ldots, t_L, c_k, \text{EOS}\}$. Here, $t_l|_{l=1}^{L}$ and $c_k$ are the word embeddings corresponding to the text template and the class label, respectively, while SOS and EOS are the start and end token embeddings. The text encoder $\mathcal{F}_t$ encodes $\tilde{Y}$ via transformer blocks to produce the latent textual feature as $\tilde{f}_t = \mathcal{F}_t(\tilde{Y}_0, \theta_t)$. For zero-shot inference, each text feature with class labels $y = \{1, 2, \cdots, C\}$ is paired with the image feature to compute a similarity score $s_i = \text{sim}(\tilde{f}_t \cdot \tilde{f}_v)$, where $\text{sim}(.)$ denotes the cosine similarity. The prediction probability on $X$ can be denoted by $p(y_i|X) = \frac{\exp(\text{sim}(\tilde{f}_t \cdot \tilde{f}_v)\tau)}{\sum_{i=1}^{K} \exp(\text{sim}(\tilde{f}_t \cdot \tilde{f}_v)\tau)}$, where $\tau$ is the temperature of the softmax.

**Prompt tuning on downstream tasks.** CLIP contains a plethora of knowledge leveraged from training on millions of noisy image-text pairs. To effectively extract the rich features learned by the CLIP model, recent approaches [48, 47, 18, 2, 42] append extra learnable prompts while keeping image and text encoders frozen. These prompts modify the context of the model input without distorting the pre-trained CLIP features. Prompts are appended either at the image or text encoder side and learn contextual information tailored towards a specific task. In our work, we use a recently introduced multi-modal prompting baseline [18] that learns prompt tokens on both the text and image encoders.

Specifically, we append learnable $V$ visual and $T$ text prompts given as $p_v = \{p_v^1, p_v^2, \ldots, p_v^V\}$, and $p_t = \{p_t^1, p_t^2, \ldots, p_t^T\}$ with the visual and textual input tokens respectively. The image encoder processes the input $\tilde{X}_0^p = \{\text{CLS}, p_v, e_1, e_2, \ldots, e_M\}$ to generate a prompted visual feature representation denoted as $\tilde{f}_v^p$. Similarly, the text encoder processes the input $\tilde{Y}_0^p = \{\text{SOS}, p_t, t_1, t_2, \ldots, t_L, c_k, \text{EOS}\}$ producing the textual feature representation $\tilde{f}_t^p$. Our approach uses deep prompting as utilized in [18], along with text prompts and conditional image prompts at subsequent transformer blocks. We jointly represent the visual and textual prompts by $p$. We refer the readers to [18] for more details on the baseline architecture.

**Test-time prompt tuning.** Test-time prompt tuning introduced by [34] aims to benefit from the rich knowledge of CLIP to boost its generalization in a zero-shot manner. TPT can be viewed as a means to provide the model with a context that is customized for each individual test sample in order to more accurately recall the knowledge contained within CLIP. During inference, several randomly augmented views are generated from the given test sample $X_{\text{test}}$. The predictions having entropy below a certain threshold are kept, while other views are discarded using a confidence selection filter. The averaged entropy of the filtered predictions is then used to update the prompts $p$ in an unsupervised fashion using the following objective function.

$$\mathcal{L}_{\text{entropy}} = \arg\min_{p} -\sum_{i=1}^{C} \tilde{p}_p(y_i|X_{\text{test}}) \log \tilde{p}_p(y_i|X_{\text{test}}), \tag{1}$$

where $\tilde{p}_p(y_i|X_{\text{test}})$ represents the mean of vector class probabilities produced by the model across the different augmented views preserved after the confidence selection filter.

### 3.2 PromptAlign

Unimodal Test-time Prompt Tuning (TPT) [34] updates the text prompts on the fly at inference by minimizing the entropy loss. This approach does not explicitly handle distribution shift that arises in the test set which is sub-optimal. One solution to this problem is to align the source and target distributions by bringing the test sample into the domain of the source data distribution. However, TPT updates the prompts only on the text encoder branch with static labels which poses an architectural limitation in performing the distribution alignment of tokens. Hence, distribution alignment of tokens

can only be performed on the vision branch. We, therefore, utilize multi-modal prompt learning VL models [18] to explicitly handle the distribution shift of each test sample from the source distribution.

Given a test sample $X_{\text{test}}$, we take multiple augmented views and pass them through the visual encoder with deep learnable prompts as shown in Fig. 2. At each layer of the visual encoder, we compute the token alignment loss between the means and variances of the test sample with that of the means and variances of a proxy dataset which mimics the behavior of the source data distribution. Our final objective combines the entropy and alignment losses to update the prompts for a given test sample.

**Proxy source dataset.** In order to compute the token embedding statistics on the source dataset, we require the pre-training dataset of the CLIP model. The CLIP model was trained on over 400 million image-text pairs, which is not publicly available. However, previous works have shown that LAION400M [33] can be used as a training dataset to achieve the performance of CLIP [5], leading to subsets of LAION400M being used as a proxy dataset for the CLIP training dataset. In addition to this, CLIP has also been heavily tuned to achieve excellent zero-shot performance on ImageNet [1]. Therefore, we use ImageNet as the proxy source dataset for computing the mean and variance of token distributions. Source dataset statistics are computed offline and utilized directly at test time.

**Token Distribution Alignment via multi-modal prompting.** We generate $N_k$ random views of the test sample using a set of augmentations $\mathcal{H}$. The mean and variance statistics of token embeddings of the test sample are computed at the output of each transformer layer of the CLIP model's visual encoder, across the $N_k$ views. Similarly, the source data statistics are pre-computed in an offline manner. We represent test sample distribution by $(\mathcal{T})$ and source distribution by $(\mathcal{D})$. Specifically, we compute the token means and variances for the alignment as follows.

$$\boldsymbol{\mu}_l(\mathcal{T}; \boldsymbol{p}) = \frac{1}{N_k} \sum_{\text{x} \in \mathcal{H}(X)} \tilde{\boldsymbol{X}}_{l,\text{x}}^{\boldsymbol{p}} \quad , \tag{2}$$

$$\boldsymbol{\sigma}_l^2(\mathcal{T}; \boldsymbol{p}) = \frac{1}{N_k} \sum_{\text{x} \in \mathcal{H}(X)} \left( \tilde{\boldsymbol{X}}_{l,\text{x}}^{\boldsymbol{p}} - \boldsymbol{\mu}_l(\mathcal{T}; \boldsymbol{p}) \right)^2 , \tag{3}$$

where $\boldsymbol{\mu}_l(\mathcal{T}; \boldsymbol{p})$ and $\boldsymbol{\sigma}_l^2(\mathcal{T}; \boldsymbol{p})$ are the vector means and variances of the test sample tokens at the layer $l$ in the visual encoder and $\tilde{\boldsymbol{X}}_{l,\text{x}}^{\boldsymbol{p}}$ represents the prompted token embeddings at layer $l$ for the augmented view input x. Similarly for each layer $l$ in the visual encoder, we pre-compute the source data statistics as,

$$\hat{\boldsymbol{\mu}}_l = \boldsymbol{\mu}_l(\mathcal{D}, \theta_v) \quad \text{and} \quad \hat{\boldsymbol{\sigma}}_l^2 = \boldsymbol{\sigma}_l^2(\mathcal{D}, \theta_v) \quad , \tag{4}$$

where $\theta_v$ denotes the parameters of the visual encoder from the pre-trained CLIP model. We compute the token distribution alignment loss between the mean and variances of the test sample and the source dataset statistics as follows,

$$\mathcal{L}_{\text{align}} = \frac{1}{L} \sum_{l=1}^{L} \left( \|\boldsymbol{\mu}_l(\mathcal{T}; \boldsymbol{p}) - \hat{\boldsymbol{\mu}}_l\|_1 + \|\boldsymbol{\sigma}_l^2(\mathcal{T}; \boldsymbol{p}) - \hat{\boldsymbol{\sigma}}_l^2\|_1 \right). \tag{5}$$

As shown above, we use $L_1$ loss to enforce the distribution alignment of the test sample with the source distribution. The alignment loss $\mathcal{L}_{\text{align}}$ is added to the entropy loss (Eq. 1) to obtain the final objective $\mathcal{L}_{\text{final}}$. Finally, the combined objective is optimized to update the prompts $\boldsymbol{p}$.

$$\mathcal{L}_{\text{final}} = \mathcal{L}_{\text{entropy}} + \beta \mathcal{L}_{\text{align}} \quad , \tag{6}$$

where the hyperparameter $\beta$ controls the contribution of alignment loss to the overall objective function. Our alignment loss in Eq. 6 bridges the gap between source and token distributions by updating the prompts to align them with the source distribution.

**Discussion on $\mathcal{L}_{\text{final}}$.** As derived above, our test-time loss combines the entropy minimization ($\mathcal{L}_{\text{entropy}}$) and distribution alignment objectives ($\mathcal{L}_{\text{align}}$) together. The $\mathcal{L}_{\text{entropy}}$ objective enforces prediction consistency among different views of a sample, leading to robustness against various variations that could possibly occur at test time. On the other hand, $\mathcal{L}_{\text{align}}$ effectively narrows the domain shift and brings the test sample distribution closer to the pre-trained CLIP distribution space, which enforces CLIP to better understand the test sample. The combination of these loss objectives harmonically adapts CLIP to be robust to different sample variations and at the same time, enhances CLIP's understanding of the underlying test sample domain for better generalization.

Table 1: **Effect of token distribution alignment strategy for domain generalization.** The base model MaPLe is trained on ImageNet and evaluated on datasets with domain shifts.

|  | Imagenet V2 | Imagenet Sketch | Imagenet A | Imagenet R | OOD Avg. |
|---|---|---|---|---|---|
| MaPLe | 64.07 | 49.15 | 50.90 | 76.98 | 60.28 |
| MaPLe+TPT | 64.87 | 48.16 | 58.08 | 78.12 | 62.31 |
| PromptAlign | **65.29** | **50.23** | **59.37** | **79.33** | **63.55** |

## 4 Experiments

**Datasets.** For the domain generalization setting, we evaluate the four out-of-distribution (OOD) variants of ImageNet [7]; ImageNetV2 [31], ImageNet-Sketch [38], ImageNet-A [16] and ImageNet-R [15]. We also evaluate the domain generalization setting on the recent challenging Photorealistic Unreal Graphics (PUG) dataset [3], comprising different textures, backgrounds, sizes and orientations. In the cross-dataset evaluation, we follow TPT [34] and evaluate the performance of methods on 10 image classification datasets covering a wide range of visual recognition tasks. This includes one generic-objects dataset Caltech101 [10]; five fine-grained datasets OxfordPets [28], StanfordCars [20], Flowers102 [27], Food101 [4] and FGVC-Aircraft [24], which contain images of animals, flowers and transportation; and four datasets of scenes, textures, satellite imagery and human actions – SUN397 [36], DTD [6], EUROSAT [13] and UCF101 [35] respectively.

**Baselines.** We evaluate PromptAlign with existing few-shot prompt learning methods for adapting CLIP including CoOp [48] and CoCoOp [47], and TPT [34] method. MaPLe [18] is a multi-modal prompt learning baseline, which adapts CLIP by learning deep prompts on both the text and vision branches. TPT is a test-time prompt tuning method, that tunes the prompt at test time per input sample achieving state-of-the-art performance in prompt learning when combined with CoOp.

**Implementation details.** Following MaPLe [18], we train on ImageNet using 16-shot training data with 2 prompt tokens for a depth of 3 layers. We optimize the prompts on both the text and vision branches using a single test image. We obtain 63 augmented views using random resized crops and horizontal flip augmentations to construct a batch of 64 images including the original image to mimic the setting of TPT. From the 64 predictions, we obtain top $10\%$ confident predictions based on the lowest entropy and compute the average prediction probability. We compute the token distribution alignment loss between the tokens of all 64 images. We optimize the prompts to minimize the combined loss of average prediction entropy and the token distribution alignment loss using the AdamW optimizer. We use a learning rate of $5e^{-4}$ for the fine-grained datasets Flowers102, OxfordPets, Food101, SUN397, FGVCAircraft, and EuroSAT and a learning rate of $0.04$ for the rest of the datasets, and set the loss scale factor $\beta$ equal to $100$. Further details are listed in Appendix A.

### 4.1 Token distribution alignment in V-L models

We first evaluate the performance of the token distribution alignment strategy over the baseline for test-time prompt tuning. Since the alignment can only be performed on the vision branch, we explicitly compare PromptAlign to vanilla MaPLe, and MaPLe with TPT using entropy minimization. Table 1 presents the results in domain generalization across ImageNet variants. TPT improves the performance of MaPLe from $60.28\%$ to $62.31\%$ on average, while the incorporation of token distribution alignment further improves the performance by $1.24\%$. This indicates that explicitly aligning the distribution of training and testing data enhances CLIP's generalization. Moreover, TPT is intended to improve test-time adaptation as a plug-in module, but its effectiveness is not uniform, as observed from the drop in performance on the ImageNet-Sketch dataset. On the other hand, PromptAlign does not show any performance degradation and improves beyond vanilla MaPLe consistently on all datasets. We further present qualitative results of PromptAlign in Appendix C.

### 4.2 Domain Generalization

Compared to CLIP, all test-time adaptation methods demonstrate better performance, indicating that V-L models benefit from test-time adaptation approaches (Table 2). PromptAlign achieves the highest Top-1 accuracy averaged across all ImageNet variants. In comparison to the previous state-of-the-art CoOp+TPT, PromptAlign achieves an average improvement of $0.71\%$. Out of the four different

Table 2: **Comparison of PromptAlign in domain generalization setting.** Prompt learning methods are trained on ImageNet and evaluated on datasets with domain shifts.

| | Imagenet V2 | Imagenet Sketch | Imagenet A | Imagenet R | OOD Avg. |
|---|---|---|---|---|---|
| CLIP [30] | 60.86 | 46.09 | 47.87 | 73.98 | 57.20 |
| CLIP+TPT [34] | 64.35 | 47.94 | 54.77 | 77.06 | 60.81 |
| CoOp [48] | 64.20 | 47.99 | 49.71 | 75.21 | 59.28 |
| CoOp+TPT [34] | **66.83** | 49.29 | 57.95 | 77.27 | 62.84 |
| Co-CoOp [47] | 64.07 | 48.75 | 50.63 | 76.18 | 59.91 |
| Co-CoOp+TPT [34] | 64.85 | 48.27 | 58.47 | 78.65 | 62.61 |
| PromptAlign | 65.29 | **50.23** | **59.37** | **79.33** | **63.55** |

Table 3: **Effect of token distribution alignment strategy for domain generalization.** The base model MaPLe is trained on ImageNet and evaluated on PUG-ImageNet.

| | Camera (Yaw/ Pitch/ Roll) | Pose (Yaw/ Pitch/ Roll) | Scale | Texture | Lighting | Worlds |
|---|---|---|---|---|---|---|
| MaPLe [18] | 48.73/ 39.93/ 32.13 | 48.10/ 28.40/ 27.80 | 46.90 | 37.90 | 15.50 | 32.13 |
| MaPLe+TPT | 57.04/ 45.99/ 39.23 | 56.26/ 35.64/ 33.26 | 54.87 | 43.73 | 22.52 | 42.00 |
| PromptAlign | **58.14/ 46.93/ 40.45** | **57.43/ 36.31/ 34.32** | **56.18** | **44.97** | **23.06** | **43.24** |

Table 4: **Comparison of PromptAlign in cross-dataset evaluation.** Prompt learning methods are trained on ImageNet and evaluated on cross-datasets.

| | Caltech | Pets | Cars | Flowers | Food101 | Aircraft | SUN397 | DTD | EuroSAT | UCF101 | *Average* |
|---|---|---|---|---|---|---|---|---|---|---|---|
| CLIP [30] | 93.35 | 88.25 | 65.48 | 67.44 | 83.65 | 23.67 | 62.59 | 44.27 | 42.01 | 65.13 | 63.58 |
| CLIP+TPT [34] | **94.16** | 87.79 | 66.87 | 68.98 | 84.67 | 24.78 | 65.50 | **47.75** | 42.44 | 68.04 | 65.10 |
| CoOp [48] | 93.70 | 89.14 | 64.51 | 68.71 | 85.30 | 18.47 | 64.15 | 41.92 | 46.39 | 66.55 | 63.88 |
| CoCoOp [47] | 93.79 | 90.46 | 64.90 | 70.85 | 83.97 | 22.29 | 66.89 | 45.45 | 39.23 | 68.44 | 64.63 |
| ProDA [46] | 86.70 | 88.20 | 60.10 | 77.50 | 80.80 | 22.20 | - | 50.90 | 58.50 | - | 65.62 |
| MaPLe | 93.53 | 90.49 | 65.57 | 72.23 | 86.20 | 24.74 | 67.01 | 46.49 | **48.06** | 68.69 | 66.30 |
| MaPLe+TPT | 93.59 | 90.72 | 66.50 | 72.37 | 86.64 | 24.70 | **67.54** | 45.87 | 47.80 | 69.19 | 66.50 |
| PromptAlign | 94.01 | **90.76** | **68.50** | **72.39** | **86.65** | **24.80** | **67.54** | 47.24 | 47.86 | **69.47** | **66.92** |

domains, our method improves beyond the previous best, except in Imagenet-V2. We hypothesize this is due to the extensive training of CoOp on ImageNet which has a very similar distribution to ImageNet-V2. We further evaluate the effectiveness of our method in domain generalization on the challenging PUG dataset as reported in Table 3. PromptAlign consistently improves upon TPT here as well. These results demonstrate the effectiveness of PromptAlign, which adapts CLIP simultaneously for both robustness against view changes and alignment of train-test set distribution.

## 4.3 Cross-Dataset Transfer

In Table 4, we evaluate the performance of our method in generalizing across diverse cross-datasets in comparison with existing state-of-the-art methods using prompt learning. PromptAlign provides consistent improvements and outperforms the previous best method MaPLe combined with TPT in all cross-datasets with an average improvement of 0.42%. In comparison to prompt learning methods, it can be observed that both CoOp and CoCoOp on average are inferior to zero-shot CLIP+TPT, while PromptAlign consistently demonstrates improvements, which validates the necessity of token distribution alignment for preserving CLIP's generalizability. This further affirms that PromptAlign harmonically combines the effect of entropy minimization and token distribution alignment in V-L models. We also analyze the effect of a better proxy source dataset, using a subset of LAION400M in Appendix D.1, proving the effectiveness of distribution alignment with source dataset statistics. Furthermore, we note that as opposed to our method, existing prompt learning methods with TPT are not consistent across cross-datasets, which we discuss in detail in Appendix B.

| Method | Entropy loss | Distribution alignment | Top-1 Acc. |
|---|:---:|:---:|:---:|
| MaPLe | ✗ | ✗ | 50.90 |
| MaPLe+TPT | ✓ | ✗ | 58.08 |
| PromptAlign$^\dagger$ | ✗ | ✓ | 50.85 |
| PromptAlign | ✓ | ✓ | **59.37** |

Table 5: **Analysis on the alignment and entropy minimization loss.** The average of Top-1 accuracy (%) across three seeds is reported. PromptAlign$^\dagger$ denotes our method excluding the entropy minimization loss.

## 5 Ablation

We perform ablative analysis on various components of PromptAlign. Unless stated otherwise, we show ablations on ImageNet-A dataset, the smallest domain generalization variant for simplicity. Detailed ablation trends across other ImageNet variants and datasets are presented in Appendix D.

**Token Distribution Alignment.** Table 5 summarizes the effect of token distribution alignment in PromptAlign. It can be observed that removing the entropy loss results in almost the same performance as vanilla MaPLe. Whereas combining alignment and entropy losses improves the performance remarkably. This is because the distribution alignment loss is a regularizer that functions together with the self-entropy loss. Since distribution alignment does not promote any discriminative learning as opposed to entropy loss, it is not expected to improve test-time adaptation on its own.

**Loss variants for distribution alignment.** We show the variation in performance with different loss choices for distribution alignment objective. Figure 4 compares the three loss choices $L_1$, $L_2$, and KL divergence. We observe that $L_1$ constraint performs best for the distribution alignment objective. Further analysis is shown in Table E of supplementary material.

**Loss balancing factor $\beta$.** We ablate the loss scale factor $\beta$ to analyze the significance of the distribution alignment loss. The performance improves over the baseline model with TPT as depicted in Figure 3, even with a smaller weight scale and gradually plateaus close to a scale of 100. We use ImageNet validation set for this ablation and choose $\beta = 100$ across all experiments. We find a similar trend of the scaling factor on other datasets, which is discussed in Appendix D. For the MaPLe+TPT model, the value of $\beta$ is 0 as there is no alignment objective in this case.

**Higher Order Statistics.** Since we align distributions computed using a small set of samples, we use relatively simpler (mean and variance) measures for our alignment strategy over higher order statistical measures. We evaluate this choice for the distribution alignment in PromptAlign using the $5^{\text{th}}$ order statistics of Central Moment Discrepancy measure [43] in Table 6. Our observation reveals that the utilization of higher order statistics results in only a slight improvement implies that there is minimal distinction between opting for higher order statistics validating mean and variance based approach as a suitable measure for alignment in our case.

**Prompt Regularization.** We compare PromptAlign to a naive prompt regularization method in a continuous test-time adaptation setting, in which the model gradually adapts to the incoming distribution. We regularize the prompts in relation to the previous prompts, denoted by PromptReg in Table 7. We also lower the learning rate to $1e^{-5}$ to prevent diverging in the continuous setting. We

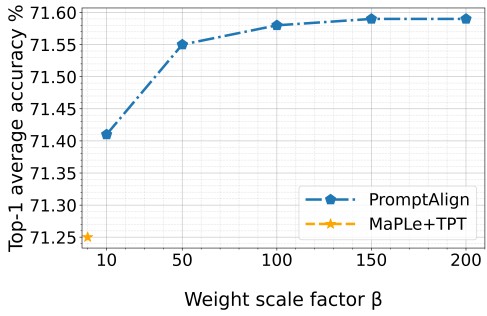

Figure 3: Effect of the loss scaling factor $\beta$ on ImageNet. Scale factor plateaus after $\beta = 100$.

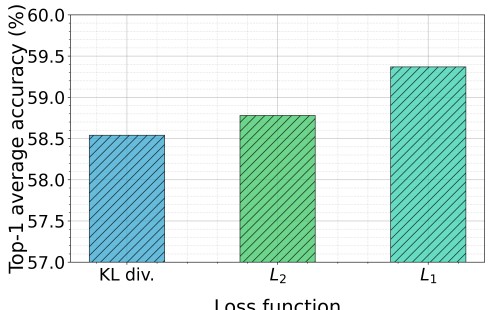

Figure 4: Effect of performance on the choice of loss function for distribution alignment.

show that the naive regularization setting, even in a continuous setting is not robust as the proposed alignment strategy, which solely aligns based on a single test sample, individually.

Table 6: **Comparison of the effect of higher order statistics for distribution alignment.** PromptAlign[†] utilizes Central Moment Discrepancy measure for the distribution alignment loss.

| | Caltech | Pets | Cars | Flowers | Food101 | Aircraft | SUN397 | DTD | EuroSAT | UCF101 | Average |
|---|---|---|---|---|---|---|---|---|---|---|---|
| PromptAlign | **72.39** | **47.24** | **90.76** | **68.5** | 69.47 | 94.01 | **86.65** | 67.54 | **24.8** | 47.86 | 66.92 |
| PromptAlign[†] | 72.36 | **47.24** | 90.74 | 68.3 | **69.48** | **94.04** | **86.65** | **67.85** | 24.75 | **48.5** | **66.99** |

Table 7: **Comparison of PromptAlign with naive prompt regularization.**

| | Target | | | | | | | | | | |
|---|---|---|---|---|---|---|---|---|---|---|---|
| | Caltech | Pets | Cars | Flowers | Food101 | Aircraft | SUN397 | DTD | EuroSAT | UCF101 | Average |
| MaPLe | 93.53 | 90.49 | 65.57 | 72.23 | 86.20 | 24.74 | 67.01 | 46.49 | **48.06** | 68.69 | 66.30 |
| PromptReg | 93.71 | 90.60 | 65.02 | 72.05 | 86.05 | 22.32 | 65.46 | 46.24 | 19.03 | 68.23 | 62.87 |
| PromptAlign | **94.01** | **90.76** | **68.50** | **72.39** | **86.65** | **24.80** | **67.54** | **47.24** | 47.86 | **69.47** | **66.92** |

**Trade-off between compute resources and performance.** Figure 5(a) demonstrates the improvement in performance as the number of augmented views at test time increases. PromptAlign shows better improvement with more augmented views, owing to a better approximation of the test sample distribution. While MaPLe+TPT performance plateaus around 64 augmented views, our method with alignment loss shows further scope for improvement given more augmented views. Figure 5(b) shows the variation in performance with the number of prompt update steps. We note that, with more update steps, PromptAlign can consistently better adapt to the test sample in comparison to MaPLe+TPT alone. In Figure 5(c) we analyze the computational overhead of PromptAlign. PromptAlign shows no additional compute cost in terms of latency. The average inference time per sample across three seeds with and without the distribution alignment strategy are $0.216s$ and $0.197s$. Considering the memory usage and increase in latency, we use $64$ views and a single-step update following TPT [34].

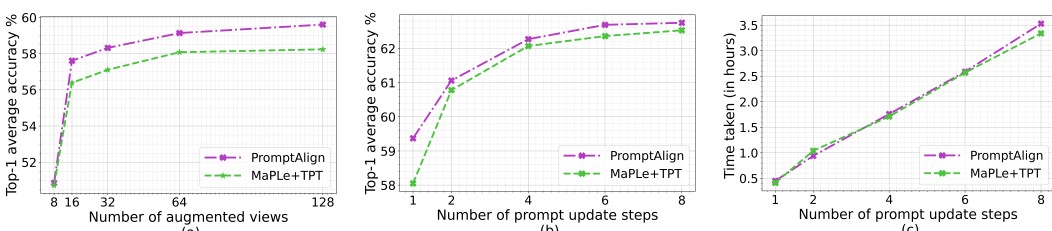

Figure 5: **Analysis of compute resource constraints on performance.** (a) The Top-1 accuracy increases with the number of augmented views. (b) The Top-1 accuracy improves consistently with the number of prompt update steps. (c) Impact on latency with the number of prompt update steps is similar for both methods.

## 6 Conclusion

In this paper, we introduce PromptAlign, a novel approach for enhancing test-time adaptation of Vision-Language (V-L) models for zero-shot generalization. Our proposed approach bridges the gap between the test sample and source distributions by explicitly aligning the test sample statistics with that of the source data distribution through token distribution alignment. To achieve this, we incorporate multi-modal prompting to facilitate the alignment of token distributions across the transformer layers during test time. Through extensive experiments, PromptAlign demonstrates superior performance over existing state-of-the-art CLIP zero-shot generalization methods in domain generalization and cross-dataset evaluation settings.

## Acknowledgements

The computational resources were provided by the National Academic Infrastructure for Super-computing in Sweden (NAISS), partially funded by the Swedish Research Council through grant agreement No. 2022-06725, and by the Berzelius resource, provided by Knut and Alice Wallenberg Foundation at the National Supercomputer Center.

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

# Appendix

In the following, we provide further details on our implementation under Section A. In Section B, we provide additional results of our proposed PromptAlign method on cross-dataset generalization, base-to-novel generalization and for different CLIP backbone architectures; and Section C shows additional qualitative results. Finally, Section D presents ablation trends across diverse datasets for the effect of the proxy dataset, the effect of test sample statistics and different parameter choices.

## A   Additional Implementation details

### A.1   Hardware and Software details

We implement our method with the MaPLe [18] multi-modal prompting model with the CLIP ViT-B/16 backbone architecture. Our models were implemented on a single NVIDIA A100 40GB GPU using the PyTorch framework.

### A.2   Effect of layer depth for token distribution alignment loss

The token distribution alignment loss is computed from all the corresponding tokens of 64 crops spanning the first 3 layers of the visual encoder. This is chosen in line with the baseline multimodal prompt learning model [18] where the deep prompts are present in layers 1-3. In order to analyze the impact of different layer depths on the alignment loss, we conduct an ablation study. Figure 6 displays the performance comparison of alignment losses obtained from various layer combinations using the ImageNet-A dataset. As observed in the Figure 6, enforcing the token alignment loss across the first three layers of the vision encoder of CLIP provides the best performance.

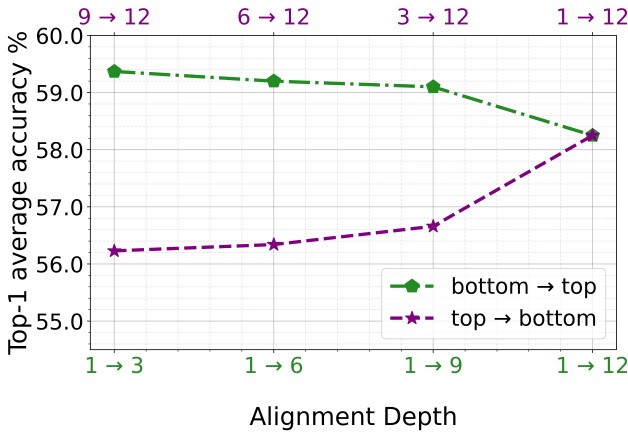

Figure 6: **Effect of alignment depth on performance**: We examine how the performance of PromptAlign is affected by altering the number of alignment layers that contribute to the alignment loss. Our findings indicate that the token distribution alignment loss is most impactful when applied to the lower layers (specifically, layer $1 \rightarrow 3$). We hypothesize that the effectiveness of the token distribution alignment loss in these layers can be attributed to the presence of our prompts, which are located within the first three layers of the visual encoder.

## B   Additional results

### B.1   Cross dataset evaluations

In Table 4 of the main paper, we compared the performance of PromptAlign with the current state-of-the-art prompt learning methods, CoOp and CoCoOp, for cross-dataset generalization. Additionally, in Table 8, we evaluate existing prompt learning methods that have been integrated with TPT and present further comparisons of our method with them. As shown in Table 8, PromptAlign outperforms both CoOp+TPT and CoCoOp+TPT in 9 out of 10 datasets. It is worth noting that when integrated with TPT, the performance of CoOp and CoCoOp decreases for the majority of cases. In contrast, PromptAlign consistently improves performance across different datasets.

### B.2   Base to Novel Generalization

We also evaluate the performance of our method on the base-to-novel generalization benchmark setting. On average, our distribution alignment strategy improves the model performance by nearly 1% in the base classes and by 0.79% on novel classes. We observe that TPT alone drops the performance of the model in some instances such as for OxfordPets, Eurosat and UCF101. On such instances, PromptAlign regulates the drop in accuracy cause by TPT, while in most cases achieves better performance in novel classes of these datasets.

Table 8: **Comparison of PromptAlign to vanilla TPT in cross-dataset evaluation.** Prompt learning methods are trained on ImageNet and evaluated on cross-datasets at test-time.

| | | | | | Target | | | | | | |
|---|---|---|---|---|---|---|---|---|---|---|---|
| | Caltech101 | OxfordPets | StanfordCars | Flowers102 | Food101 | Aircraft | SUN397 | DTD | EuroSAT | UCF101 | *Average* |
| CoOp [48] | 93.70 | 89.14 | 64.51 | 68.71 | 85.30 | 18.47 | 64.15 | 41.92 | 46.39 | 66.55 | 63.88 |
| CoOp+TPT [34] | 93.15 | 89.48 | 66.77 | 68.48 | 86.48 | 20.51 | 66.06 | 43.32 | 37.73 | 68.91 | 64.08 |
| CoCoOp [47] | 93.79 | 90.46 | 64.90 | 70.85 | 83.97 | 22.29 | 66.89 | 45.45 | 39.23 | 68.44 | 64.63 |
| CoCoOp+TPT [34] | 88.57 | 85.33 | 59.68 | 55.31 | 80.64 | 16.89 | 60.24 | 38.93 | **48.55** | 63.35 | 59.75 |
| PromptAlign | **94.01** | **90.76** | **68.50** | **72.39** | **86.65** | **24.80** | **67.54** | **47.24** | 47.86 | **69.47** | **66.92** |

Table 9: **Comparison on Base-to-novel generalization of PromptAlign with previous methods.** PromptAlign shows consistent improvement over TPT.

| Dataset | | ProDA [46] | MaPLe [18] | MaPLe+TPT [34] | PromptAlign (Ours) |
|---|---|---|---|---|---|
| Avg. on 11 datasets | Base | 81.56 | 82.24 | 82.16 | **83.19 (+0.95)** |
| | Novel | 72.30 | 75.09 | 74.95 | **75.88 (+0.79)** |
| ImageNet | Base | 75.40 | 76.67 | 77.73 | **78.26** |
| | Novel | 70.23 | 70.54 | 72.24 | **72.59** |
| Caltech101 | Base | 98.27 | 98.00 | 98.54 | **98.60** |
| | Novel | 93.23 | 94.27 | 94.29 | **94.50** |
| OxfordPets | Base | **95.43** | **95.43** | 95.23 | 95.38 |
| | Novel | **97.83** | 97.80 | 97.37 | 97.56 |
| Stanford Cars | Base | 74.70 | 72.90 | 74.00 | **75.02** |
| | Novel | 71.20 | 73.97 | 75.20 | **75.71** |
| Flowers102 | Base | **97.70** | 95.93 | 96.24 | 96.61 |
| | Novel | 68.68 | **72.40** | 72.10 | 72.34 |
| Food101 | Base | 90.30 | 90.70 | 91.13 | **91.63** |
| | Novel | 88.57 | 92.07 | 92.03 | **92.68** |
| FGVC Aircraft | Base | 36.90 | **37.27** | 34.31 | 37.21 |
| | Novel | 34.13 | 35.53 | 35.81 | **37.27** |
| SUN397 | Base | 78.67 | 80.80 | 81.15 | **81.57** |
| | Novel | 76.93 | 78.70 | 79.18 | **79.48** |
| DTD | Base | 80.67 | 80.30 | 82.20 | **82.60** |
| | Novel | 56.48 | 59.23 | 59.91 | **60.55** |
| Eurosat | Base | 83.90 | 93.63 | 91.02 | **94.10** |
| | Novel | 66.00 | **72.87** | 68.96 | 72.71 |
| UCF101 | Base | **85.23** | 82.97 | 82.23 | 84.11 |
| | Novel | 71.97 | 78.57 | 77.34 | **79.30** |

## B.3 Different CLIP backbone architectures

We also evaluate our method on the ViT-B/32 backbone architecture variant of CLIP. Here, we train CLIP ViT-B/32 in the original MaPLe setting without any hyperparameter tuning specific to MaPLe. Our distribution alignment strategy improves upon TPT on average in both the domain generalization setting and cross-dataset settings as reported in Table 10 and Table 11.

Table 10: **Comparison of PromptAlign in domain generalization setting for ViT-B/32.** Prompt learning methods are trained on ImageNet and evaluated on datasets with domain shifts.

| | Imagenet V2 | Imagenet Sketch | Imagenet A | Imagenet R | OOD Avg. |
|---|---|---|---|---|---|
| MaPLe | 57.63 | 42.15 | 32.12 | 67.64 | 49.89 |
| MaPLe+TPT | 60.01 | 43.77 | 37.52 | 71.11 | 53.10 |
| PromptAlign | **60.43** | **44.24** | **38.02** | **71.44** | **53.53** |

Table 11: **Comparison of PromptAlign in cross-dataset evaluation on ViT-B/32.** Prompt learning methods are trained on ImageNet and evaluated on cross-datasets.

| | Caltech | Pets | Cars | Flowers | Food101 | Aircraft | SUN397 | DTD | EuroSAT | UCF101 | *Average* |
|---|---|---|---|---|---|---|---|---|---|---|---|
| MaPLe [18] | **92.50** | 88.13 | 59.93 | 65.33 | 81.00 | 17.53 | 65.00 | 41.70 | **40.80** | 63.63 | 61.56 |
| MaPLe+TPT [34] | 91.44 | **88.47** | 59.35 | 66.08 | **82.08** | 18.71 | 66.07 | 40.01 | 39.67 | 64.40 | 61.63 |
| PromptAlign | 92.10 | 88.44 | **63.48** | **66.14** | 82.07 | **18.76** | **66.08** | **42.54** | 39.68 | **65.57** | **62.49** |

# C Qualitative Results

We generate attention map visualizations from the heads of the last block of the CLIP visual encoder on the test samples from the Imagenet-A dataset. In Figure 7, we compare the attention maps produced by PromptAlign and TPT. Our findings reveal that PromptAlign exhibits a stronger focus on discriminative regions while reducing attention on background regions compared to TPT. For the first test sample (on the left), PromptAlign intensifies attention on the object of interest and diminishes attention on the background regions. Similarly, in the second image (on the right), PromptAlign increases attention toward the salient object.

# D Extended Ablations

## D.1 Proxy source dataset

We assess the effectiveness of our alignment strategy, in Table 12 by employing LAION400M [33] to compute the source data statistics. We utilize a subset of 2 Million images[*] from the LAION400M dataset (denoted as PromptAlign[†]). We show that using LAION400M to compute the source dataset statistics enhances the distribution alignment objective, and thus the overall performance in Table 12. This emphasises the effectiveness of distribution alignment, since LAION400M has been shown as a valid training dataset for CLIP [5].

In Table 13 we present the performance of our method when using the training set of the respective dataset as the proxy source data to compute the training statistics used for the token distribution alignment loss. We choose 5 datasets: DTD [6], StanfordCars [20], UCF10 1[35] Caltech101 [10], and FGVC Aircrafts [24] for the analysis. We observe that using ImageNet as the proxy source dataset gives better performance across all datasets, except in UCF101, where the performance is the same. Experiments in different Vision-Language models such as EVA-CLIP [9], with different training recipes has a similar trend validating the choice of ImageNet as a proxy dataset.

---

[*]Due to compute and time limitations we use a subset of LAION400M dataset

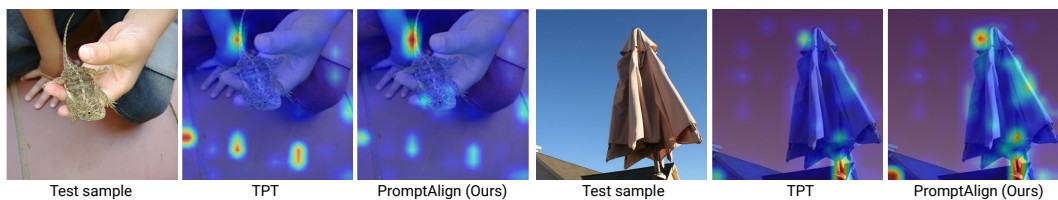

| Test sample | TPT | PromptAlign (Ours) | Test sample | TPT | PromptAlign (Ours) |

Figure 7: **Comparison of attention map visualizations.** PromptAlign focuses on highlighting the discriminative areas while reducing attention on background regions as opposed to TPT.

Table 12: **Experiment with different source dataset statistics.** PromptAlign[†] refers to source dataset statistics computed using a subset of LAION400M for distribution alignment.

| | Caltech101 | OxfordPets | StanfordCars | Flowers102 | Food101 | Aircraft | SUN397 | DTD | EuroSAT | UCF101 | *Average* |
|---|---|---|---|---|---|---|---|---|---|---|---|
| PromptAlign | 94.01 | **90.76** | **68.50** | **72.39** | 86.65 | **24.80** | 67.54 | 47.24 | 47.86 | 69.47 | 66.92 |
| PromptAlign[†] | **94.32** | **90.76** | 68.11 | 72.38 | **87.26** | 24.78 | **68.02** | **47.97** | **47.88** | **70.20** | **67.17** |

Table 13: **Comparison of the effect of proxy source dataset in cross-dataset evaluation for different V-L models.** PromptAlign[‡] refers to PromptAlign with the training set of the respective test dataset used as the proxy source dataset to compute statistics for the alignment loss.

| Model | Method | Target | | | | |
|---|---|---|---|---|---|---|
| | | Caltech101 | StanfordCars | UCF101 | Aircraft | DTD |
| CLIP | PromptAlign[‡] | 93.98 | 68.44 | **69.47** | 23.85 | 47.10 |
| | PromptAlign | **94.01** | **68.50** | 69.47 | **24.80** | **47.24** |
| EVA-CLIP | PromptAlign[‡] | 95.66 | 79.44 | 65.39 | **23.57** | 50.06 |
| | PromptAlign | **95.69** | **79.50** | 65.41 | 23.48 | **50.06** |

Our hypothesis on the effectiveness of ImageNet as a proxy source dataset is two-fold. Firstly, the token distribution alignment loss intends to align the tokens to that of the CLIP pre-training dataset token distributions. The larger scale of ImageNet in comparison to the training set of the above datasets, and the CLIP pre-training strategy, finetuning it to improve performance on the ImageNet benchmark [1] renders ImageNet a better proxy source dataset for CLIP. Furthermore, using ImageNet acts as a regularizer, maintaining the generalization capability of CLIP while performing test-time prompt learning using entropy minimization. Similarly, since LAION400M is a better representation of the CLIP training dataset, the subset of LAION400M used as the proxy shows an improvement over ImageNet in being used as a proxy source dataset.

## D.2 Effect of test sample statistics

To analyze the effect of the test sample statistics quality on the distribution alignment strategy, we experiment with the use of more samples for alignment. For each test example, we use a bag of samples (N=5) obtained from the same class (while the class label is kept unknown) to estimate the test sample statistics. We present the results in Table 14 which shows that test sample statistics can be better approximated with more samples, improving the performance through the alignment objective. However, in the actual setting of test-time adaptation, only a single sample is available.

## D.3 Trend across parameter choices

We present ablations on the distribution loss scaling factor $\beta$, the number of augmented views, and the number of prompt update steps across diverse datasets. We choose 5 datasets: DTD [6], StanfordCars [20], UCF101 [35] and Caltech101 [10], and ImageNet-A [16], and compare the average accuracy across them as shown in Figure 8.

Table 14: **Evaluation of the effect of more samples for distribution alignment.** PromptAlign[*] utilizes a bag-of-samples from the same class for the distibution alignment.

| | Caltech | Pets | Cars | Flowers | Food101 | Aircraft | SUN397 | DTD | EuroSAT | UCF101 | *Average* |
|---|---|---|---|---|---|---|---|---|---|---|---|
| PromptAlign | 94.01 | 90.76 | 68.50 | 72.39 | 86.65 | 24.80 | 67.54 | 47.24 | 47.86 | 69.47 | 66.92 |
| PromptAlign[*] | **96.12** | **91.60** | **74.00** | **74.49** | **87.67** | **25.68** | **68.83** | **51.52** | **50.86** | **74.81** | **69.59** |

The loss scaling factor $\beta$ plateaus close to 100, whereas the performance increases with the increasing number of augmented views and prompt update steps. As discussed in Section 5, the token distribution alignment scaling factor $\beta$ is set to 100 using the ImageNet validation set, which is used for all experiments.

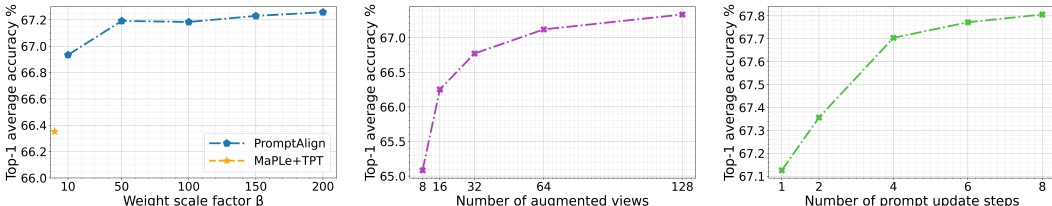

Figure 8: **Effect of loss scaling factor** $\beta$ (left)**, number of augmented views** (center)**, and number of prompt update steps** (right) **averaged across the 5 datasets.** The loss scaling factor trend is similar to that on ImageNet validation set, and the average accuracy increases consistently with the number of augmented views and the number of prompt update steps.

# E    Analysis of losses for distribution alignment

The comparison of different alignment loss functions is presented in Fig 4 of the main paper. All three losses improve on top of the test-time prompting approach. In terms of a test-time setting, the L1 loss appears better suited to equally valuing all errors as opposed to L2 which primarily penalizes outliers. Moreover, the KL divergence loss does not enforce strict distribution alignment hence leading to sub-par performance. We further show a comparative analysis of alignment loss gradients in Table 15. Each row indicates the ratio of the average gradient in comparison to the gradient when using each of the other losses. In our experiments we notice that the shift in the statistical measures are small, which would lead to very small gradients in the case of L2 and KL divergence losses. This can be observed from Table 15. Specifically, the gradient of the L1 loss is almost 80 times greater than that of the L2 loss and approximately 4 times greater than that of the KL divergence loss. It is evident that the L1 Loss results in improved alignment, primarily due to its higher gradient values, indicating that prompt updates with an L1 penalty can yield better alignment.

Table 15: **Comparison of different alignment losses.** L1 Loss yields better alignment due smaller deviation in statistical measures, thus higher gradient values.

|  | L1-Loss | L2-Loss | KL-Divergence |
|---|---|---|---|
| L1-Loss | 1 | 81.40 | 3.95 |
| L2-Loss | 0.0123 | 1 | 0.048 |
| KL-Divergence | 0.25 | 20.5 | 1 |

# F    Limitations and Future Directions

As a test-time adaptation strategy, PromptAlign improves CLIP performance but relies on the proxy source dataset statistics for aligning the train data distribution with the test sample statistics. Further, it involves a single update step during runtime to optimize the prompts. A potential future direction would be to explore the adaptation of PromptAlign to other downstream tasks such as object detection and segmentation.

