# OpenReview forum: "Align Your Prompts: Test-Time Prompting with Distribution Alignment for Zero-Shot Generalization"
_NeurIPS.cc/2023/Conference — NeurIPS 2023 poster_

### Official Review · Reviewer_JL6L · 2023-06-26

**Soundness:** 3 good
**Presentation:** 3 good
**Contribution:** 3 good
**Rating:** 5
**Confidence:** 4

**Summary:**

This paper tackles the task of test-time prompt tuning. It proposes to align the out-of-distribution test sample statistics to those of the source data. The ImageNet is used as an effective proxy source dataset. Evaluation results show that the proposed method obtains better performances on domain generalization and cross-dataset generalization tasks.

**Strengths:**

1. This paper studies test-time adaptation for foundation models. Given the popularity of V-L models and prompt tuning, the paper may be interesting to many researches.

2. Exploring distribution alignment in V-L models at test time is new. The proposed method is easy to implement and can be generalized to different model architectures.

3. The paper is well structured and easy to follow. Experiments are extensive enough. PromptAlign obtains consistently better accuracies than previous methods in most cases.

**Weaknesses:**

1. The method highly relies on the ImageNet statistics. In the supplementary, the authors show that using ImageNet statistics gives better performance than using other small datasets. It is unclear if this also applies to V-L models other than CLIP.

2. The alignment loss, in my understanding, is a regularization term on the visual prompts to avoid large devation during adaptation. A simple manner is to regularize the difference between initial prompt values (or statistics from the test image) and adapted values. Such comparison is missing.

**Questions:**

1. In Tab.4, is PromptAlign$^\dagger$ equivalent to MaPLe+alignment loss? If the alignment loss alone does not improve test-time adaptation, how to validate that narrowing the distribution gap real matters when adapting with entropy loss?

2. How does the proposed method perform on different CLIP backbone architectures?

**Limitations:**

The authors have included possible limitations in the end of paper. The discussions are constructive.

---

> ### Author Rebuttal · Authors · 2023-08-09
>
> ## **Effect of proxy source dataset on other VL models**
>
> We evaluate our method by implementing the MaPLe architecture in the recently introduced EVA-CLIP(CVPR 2023) V-L model [1]. We conduct the same experiment reported in Table 6 of the paper on the EVA-CLIP model. We show that the use of ImageNet as the training dataset to compute source statistics is still relatively better than the use of the training data statistics for distribution alignment.
>
> ### Comparison of the effect of the proxy dataset against, the training dataset of the dataset for EVA-CLIP.
>
> | Method | Caltech | Cars | UCF101 | Aircraft | DTD |
> |-----|-----|---|---|---|---|
> | TrainSet | 95.66 | 79.44 | 65.39 | 23.48  | 50.06  |
> | ImageNet | 95.69 | 79.50 | 65.41  | 23.57  | 50.06  |
>
>
> ## **Replacing PromptAlign with naive regularization technique**
> As suggested we experiment with the simpler regularization of prompts based on the previous update of the prompts. This results in a continuous test-time adaptation setting for which we choose a lower learning rate to ensure the prompts do not diverge. While effective, we notice that such a simpler regularization is not consistent and overall drops in performance in comparison to baseline MaPLe (predominantly due to the large drop in the EuroSAT dataset). Regardless, our proposed alignment strategy is better across all datasets. PromptAlign* in the below Table is the suggested naive prompt regularization method.
>
> ### Comparison against prompt regularization using continuous adaptation.
>
> | Method|Flowers|DTD|Pets|Cars|UCF|Caltech|Food|SUN   |Aircraft|Eurosat| Average |
> |-----|-----|---|---|---|---|-----|----|---|-----|-----|----|
> | MaPLe | 72.39  | 47.24  | 90.76  | 68.5  | 69.47  | 94.01  | 86.65  | 67.54  | 24.80  | 47.88  |66.92 |
> | PromptAlign*  | 72.05 | 46.24 | 90.60 |	65.02 |	68.23 |	93.71 |	86.05 |	65.46 |	22.32 | 19.03 | 62.87 |
> | PromptAlign  | 72.38  | 47.97  | 90.76  | 68.11  | 70.2  | 94.32 |  87.26 | 68.02  |  24.78 | 47.88 |67.17 |
>
> ## **Mutual synergy between Distribution alignment and Entropy objectives**
> Yes, PromptAlign^{dagger} in Table 4 in the paper refers to MaPLe with only the distribution alignment loss (i.e. without the entropy minimization objective). The proposed alignment strategy treats each token as a random variable, where the distribution of the source variable is matched for the test sample's distribution. This regularization works in synergy with the entropy minimization objective and improves the learning objective. Unlike entropy minimization, the alignment objective does not promote any discriminative learning, and thus cannot be solely used for test-time adaptation.
>
> ## **Different CLIP backbone architecture**
> We train MaPLe on the ViT-B/32 architecture and evaluate our method on the cross-dataset and domain generalization settings. The results are consistent, showing a similar trend in performance. We thank the reviewer for the suggestion, and we will include this in the final draft of the paper.
>
> ### Cross Dataset benchmark evaluation on ViT-B/32 backbone
>
> | Method|Flowers|DTD|Pets|Cars|UCF|Caltech|Food|SUN   |Aircraft|Eurosat| Average |
> |-----|-----|---|---|---|---|-----|----|---|-----|-----|----|
> | MaPLe | 65.33 |	41.70 |	88.13 |	59.93 |	63.63 |	92.50 |	81.00 |	65.00 |	17.53 |	40.80 |	61.56 |
> | MaPLe+TPT  | 66.08 |	40.01 |	88.47 |	59.35 |	64.40 |	91.44 |	82.08 |	66.07 |	18.71 |	39.67 |	61.63 |
> | PromptAlign  | 66.14 |	42.54 |	88.44 |	63.48 |	65.57 |	92.10 |	82.07 | 66.08 |	18.76 |	39.68 |	62.49 |
>
> ###  Domain Generalization evaluation on ViT-B/32 backbone
> | Method | ImageNet | ImageNet A | ImageNet R | ImageNet K | ImageNet V2 | OOD Average |
> |-----|-----|----|---|-----|-----|----|
> | MaPLe | 65.27 |32.12 |67.64 |42.15|	57.63 |	49.89 |
> | MaPLe+TPT  | 67.16 | 	37.52 |	71.11 |	43.77 |	60.01 |	53.10 |
> | PromptAlign  | 67.42 | 38.02 | 71.44 |44.24 | 60.43 |	53.53 |

---

> > ### Comment · Reviewer_JL6L · 2023-08-16
> >
> > Dear Authors,
> >
> > Thanks for the detailed response. I've carefully read that and other reviews.
> >
> > I appreciated the supplemented results on other VL models, naive regularization, different CLIP backbone. Those results are postive to me.

---

### Official Review · Reviewer_ftF2 · 2023-06-27

**Soundness:** 3 good
**Presentation:** 3 good
**Contribution:** 2 fair
**Rating:** 4
**Confidence:** 5

**Summary:**

Building on top of the prompting method and architecture from MaPLE and test-time adaptation strategy of TPT, the work ads an additional alignment penalty, based on pre-computed statistics. The statistics are computed offline on a target source dataset. The method is evaluated then for domain generalisation on a set of datasets.

**Strengths:**

- The paper reads well and its easy to follow.
- Table 1 reports gains compared with the TPT method baseline.
- The conept of test-time adaptation is interesting.

**Weaknesses:**

- Somewhat low novelty: The method combines 2 existing works: takes the architecture and prompting strategy from MaPLE and the adaptation mechanism from TPT. The only addition is an alignment loss based on pre-computed statistics.
- Can this approach be combined with other methods (ie. CoOp, CoCoOp etc), similarly as its the case for TPT. If so, what is its performance? Further evaluations are needed in this direction.
- Similarly, the comparison from Table 2 and 3 are unfair. The proposed method should be compared with a similar baseline, i.e. TPT+MaPLE, as done in Table 1. Currently, especially for Table 3 its unclear what are the gains, if any.
- Missing comparisons with current state-of-the-art methods (e.g. Neural Prompt Search, LASP, ProDA etc).
- No evaluation on the standard settings from CoOp (i.e. without ImageNet pretraining) and CoCoOp( base-to-new setting).
- Missing analysis of certain ablation. For example, why is L1 offering the best results in this case? What makes it more suitable?
- What is the impact of the quality of the statistics on the overall performance?
- No FLOPs/compute analysis: What is the inference time and FLOPs (i.e. including adaptation time) per sample? How does this compare with TPT? How much extra time is spent on pre-computing the statistics? It looks than in some case ity may be simply faster to train and eval CoOp or a similar method, especially as the size of the dataset and/or the number of classes increases.
- L154, sum from 1 to K, where is the index i used on the denominator? It's a bit ambigous currently which f_t abd f_v are used.

**Questions:**

See the above section

**Limitations:**

Adequate limitation statement is present.

---

> ### Author Rebuttal · Authors · 2023-08-09
>
> ## **Contributions and Novelty**
> Kindly note that the multimodal prompting architecture of MaPLe and the adaptation strategy are independently pre-existing methods. However, the holistic combination of them using the proposed distribution alignment strategy to effectively use the synergy in V-L models for test-time adaptation is novel to the best of our knowledge. Further, the proposed method consistently improves on top of the individual methods across extensive benchmarks including a newly reported base-to-novel generalization benchmark. The test-time adaptation strategy using a single sample, consistently improving across numerous benchmarks showcases the effectiveness of the proposed method.
>
> ## **Combining PromptAlign with other prompting methods**
> The existing prompt learning methods such as CoOp and CoCoOp learn prompts only on the language side.
> In the test-time adaptation setting, the prompt vectors are reset to the initial states for each incoming sample. For a given dataset, since the class names and the prompt vectors are the same, the tokens throughout the text encoder will be the same. Therefore, the mean would be the token itself and the variance zero, thus leading to no contribution from a distribution alignment loss. This architectural limitation of text-side-only prompting prevents naive incorporation of a distribution alignment. However, through multimodal prompting, the distribution alignment on the vision branch allows prompts on both branches to be updated together.
>
> ## **Main paper table comparisons**
> We update the comparisons in Table 2  (in the global rebuttal response) on the cross-dataset evaluation with MaPLe, MaPLe+TPT, and our method PromptAlign. It can be noted that the gain from our alignment strategy on top of MaPLe+TPT is relatively higher than the improvement of TPT on MaPLe. We further incorporate the performance of our method with the source dataset statistics computed using a subset of LAION400M with 2.5 Million images (2x of ImageNet), since LAION400M is closer to the training dataset of CLIP [1]. These results further validate our method where the performance improves with a better proxy source dataset.
>
> ## **Comparison to the state-of-the-art methods**
> We report comparisons to ProDA from the original paper [2] since the official code is not publicly available. The cross-dataset evaluation is reported in Table 2 and the base to novel benchmark settings, in Table 3. Both Neural Prompt Search and LASP works have not been published at the time of this submission.
>
> ## **Comparisons with the standard setting of CoOp and CoCoOp base to novel benchmark**
> We compare the standard CoOp setting without ImageNet pretraining in the cross-dataset setting (mentioned as CLIP+TPT) in Table 2. We also evaluate our method in the base to novel benchmark setting in Table 3 in comparison to the state-of-the-art method MaPLe. We show that our alignment strategy improves upon MaPLe in both base and novel class settings by +0.95 and +0.79 respectively, whereas TPT alone drops in performance slightly.
>
> ## **Analysis of matching losses for Distribution Alignment**
> The comparison of different alignment loss functions is presented in Figure 4 of the main paper. All three losses improve on top of the test-time prompting approach. In terms of a test-time setting, the L1 loss appears better suited to equally valuing all errors as opposed to L2 which primarily penalizes outliers. Moreover, the KL divergence loss does not enforce strict distribution alignment hence leading to sub-par performance. Further analysis of this will be included in the final draft of the paper.
>
> ## **Impact of statistics quality on performance**
> We evaluate the quality of statistics in terms of the pre-computed source data statistics and the test sample statistics. We use a subset of LAION400M with 2.5 Million images as the proxy source dataset to compute the offline statistics. This further improves the alignment objective (Table 2, last row), suggesting that better quality in the source data statistics improves the alignment objective further.
>
> To evaluate the effect of the quality to test sample statistics, we conduct the following experiment. We randomly sample 5 images from the same class of the test sample (while the class label is kept hidden). We use these five samples to calculate the statistics of the test sample and perform test-time adaptation. The results in Table 1 reflect that the performance increases with better quality in test-sample statistics as well.
>
> ## **Compute Analysis**
> As recommended, we report the compute cost in terms of time and FLOPS including the adaptation step, in comparison to TPT in Table 4. Figure 5(c) in the submission paper denotes the compute time averaged across three seeds for the ImageNet-A dataset, for different number of prompt update steps. In comparison to the average inference time and FLOPS per sample, our method with the alignment strategy has minimal overhead.
>
> The offline pre-computation of source data statistics requires 2 forward passes of the ImageNet training set which takes nearly 3h. First, the mean is calculated and in the second pass the variance is computed using the mean that has been computed. However, this is a single-time offline computation that need not be repeated again which does not depend on the test dataset size nor the number of classes. Furthermore, since we consider the test-time adaptation setting using a single sample, we presume no training data to be available for our adaptation.
>
> ## **Minor correction in L154**
> Thank you for pointing out this error. This has been rectified. Here, 'i' refers to the index of the class, which has been now added.
>
> [1]: "Reproducible scaling laws for contrastive language-image learning." CVPR 2023.
> [2]: "Exploring visual prompts for adapting large-scale models." arXiv preprint arXiv:2203.17274 (2022).
> [3]: "Prompt distribution learning." Proceedings of the IEEE/CVF CVPR. 2022.

---

> > ### Comment · Reviewer_ftF2 · 2023-08-21
> >
> > Thanks for providing the detailed response and appologies for the delay. Some of my concerns where addressed (suggested experiments, FLOPs analysis), however, the novelty component remains somewhat on the low side given the changes with respect to TPT and the heavy re-use of existing components. The gains on the base-to-new setting are also modest given the computational increase compare with no test time adaptation. I have upgraded my score accordingly.

---

> ### Author Response · Authors · 2023-08-18
> **We thank the reviewer again for the valuable feedback and happy to address any remaining concerns.**
>
> We extend our sincere gratitude to the reviewer for their valuable time and insightful feedback. We value your constructive feedback and hope that our responses have appropriately addressed all the concerns.
>
> We really appreciate the valuable time to respond to our feedback based on the reviewer's comments. Further, we are happy to address any remaining concerns.

---

### Official Review · Reviewer_H4M3 · 2023-07-05

**Soundness:** 3 good
**Presentation:** 3 good
**Contribution:** 2 fair
**Rating:** 5
**Confidence:** 4

**Summary:**

This paper introduces a distribution alignment strategy, termed PromptAlign, for V-L models to improve test-time adaptation. To be specific, they propose a distribution alignment loss that utilizes offline computed source data statistics to encourage the test sample token distributions to be aligned with the source data token distributions. Since CLIP-pre-training data is not publicly released, they directly use ImageNet as a possible candidate for the source distribution. Experiments on multiple benchmark datasets reveal the zero-shot generalization ability of the proposed PromptAlign during test-time adaptation.

**Strengths:**

- The paper is well-written with a clear structure. The research on prompting for large Vision-Language models holds practical value.

- The proposed PromptAlign, with its straightforward implementation, has achieved good results on a variety of test benchmarks.

**Weaknesses:**

- *Proxy source dataset*. Although using an off-the-shelf large-scale dataset for distribution alignment is intuitive. It lacks theoretical justifications or insights regarding the choice, making the whole process somewhat heuristic. Moreover, as the statistics of the source dataset are computed offline, how can the method be applied to some dynamically changing environments that may significantly differ from the source data distribution?

- *Distribution alignment*. Using mean and variance for distribution alignment still lacks theoretical or more in-depth analyses. This appears to be the foundation of the paper, yet I haven't observed a particularly robust theoretical explanation. Could the authors provide more insights regarding this point?

- *Experimental results*. Based on the reported experimental results, the proposed method does not demonstrate a significant performance improvement compared to TPT-based methods.

**Questions:**

Please refer to the weaknesses.

---

> ### Author Rebuttal · Authors · 2023-08-09
>
> ## **ImageNet as proxy source dataset and precomputed source data statistics**
>
> Given the excellent zero-shot performance of CLIP on ImageNet we choose ImageNet as a feasible proxy source dataset, which is consistent with previous work [1]. Additionally, we conduct an ablation study on the alignment strategy, employing LAION400M as the source dataset, which has been shown to reflect the training dataset of CLIP [2]. Due to time constraints, we utilize a subset of LAION400M, containing 2.5 million images (2x of ImageNet size). Notably, the performance impact of using ImageNet does not considerably change when employing this subset of LAION400M. We will include these results in the main paper.
>
> ### LAION vs. ImageNet statistics for alignment loss.
>
> | Method|Flowers|DTD|Pets|Cars|UCF|Caltech|Food|SUN   |Aircraft|Eurosat| Avg |
> |-----|-----|---|---|---|---|-----|----|---|-----|-----|----|
> | ImageNet | 72.39  | 47.24  | 90.76  | 68.5  | 69.47  | 94.01  | 86.65  | 67.54  | 24.8  | 47.88  |66.92 |
> | LAION  | 72.38  | 47.97  | 90.76  | 68.11  | 70.2  | 94.32 |  87.26 | 68.02  |  24.78 | 47.88 |67.17 |
>
> The source data statistics are computed on the proxy source data to obtain the distribution of tokens that are used for the alignment during test time. Since this proxy dataset is based on the similarity to the training set of the model, this offline computation will not change regardless of the changing environment and only needs to be computed once.
>
> ## **Distribution alignment strategy**
> Higher-order statistical alignment is a classical approach in test-time domain adaptation. However, since we align distributions computed using a small set of samples, we use the relatively simpler (mean and variance) measures for our alignment strategy. We evaluate the choice of higher-order statistical measures for the distribution alignment using the Central Moment Discrepancy measure [3]. However, as shown in the Table below, we note that the increment is minimal which validates that the mean and variance based approach is a suitable measure for alignment in our case.
>
> ### Comparison of PromptAlign vs Higher order statistics for Alignment
>
> | Method|Flowers|DTD|Pets|Cars|UCF|Caltech|Food|SUN   |Aircraft|Eurosat| Avg |
> |-----|-----|---|---|---|---|-----|----|---|-----|-----|----|
> | PromptAlign | 72.39  | 47.24  | 90.76  | 68.5  | 69.47  | 94.01  | 86.65  | 67.54  | 24.80  | 47.88  |66.92 |
> | w/ CMD  | 72.36  | 47.24  | 90.74  | 68.3  | 69.48  | 94.04 |  86.65 | 67.85  |  24.75 | 48.50 |66.99 |
>
> ## **Experimental results**
> The proposed method is a lightweight adaptation strategy for V-L models of which the improvements are also appreciated by reviewers wdgw, JL6L. Being a test-time adaptation method using a single sample, the improvements are consistent across multiple datasets and benchmarks (domain generalization and cross-datasets transfer) and improve over existing works such as [4] which performs test-time prompt tuning on a single sample.
>
> In addition, we also evaluate our method in the Base to Novel benchmark, where the model is trained on the base class training set and tested on both the test sets of base classes and novel classes. As reported in Table 3 in the global rebuttal response, averaged across 11 datasets, our method shows consistent improvements with +0.95 on the base classes and +0.79 on novel classes whereas, without the alignment strategy, the performance drops below the baseline.
>
> [1]: Bahng, Hyojin, et al. "Exploring visual prompts for adapting large-scale models." arXiv preprint arXiv:2203.17274 (2022).
>
> [2]: Cherti, Mehdi, et al. "Reproducible scaling laws for contrastive language-image learning." Proceedings of the IEEE/CVF Conference on Computer Vision and Pattern Recognition. 2023.
>
> [3]: Zellinger, Werner, et al. "Central moment discrepancy (cmd) for domain-invariant representation learning." arXiv preprint arXiv:1702.08811 (2017).
>
> [4]: Shu, Manli, et al. "Test-time prompt tuning for zero-shot generalization in vision-language models." Advances in Neural Information Processing Systems 35 (2022): 14274-14289.

---

> > ### Comment · Reviewer_H4M3 · 2023-08-17
> > **Thanks for the responses**
> >
> > I appreciate the authors' responses. My major concerns have been well addressed. After carefully reviewing the feedback from other reviewers, the authors' replies, and re-reading the original manuscript, I've decided to maintain a relatively positive score.

---

### Official Review · Reviewer_wdgw · 2023-07-08

**Soundness:** 3 good
**Presentation:** 3 good
**Contribution:** 3 good
**Rating:** 6
**Confidence:** 4

**Summary:**

In this paper, the authors proposed a novel approach for enhancing test-time prompting with distribution alignment of Vision-Language (V-L) models for zero-shot generalization. A loss between the means and variances of the test sample and the proxy source dataset statistics was proposed to align the distributions during testing. The authors conducted several experiments on multiple datasets and showed improved performance over several baseline methods.

**Strengths:**

1. Combining Test-Time Prompting and Distribution Alignment for Zero-Shot Generalization makes sense to me and the proposed alignment loss is also technically sound.
2. The authors conducted extensive experiments and ablation studies to sufficiently validate the proposed approach over multiple baseline methods on several benchmarking datasets.
3. Writing is good and easy to follow. The tables and figures are also easy to understand.

**Weaknesses:**

1. I am not sure the mean and variance of a single test example would be sufficient for distribution alignment even though multiple augmented views are used. Could this lead to over-fitting? Would there be further gain with more test examples?
2. I am not sure how the proxy source dataset would affect the performance. Since LAION400M is more closer to the training data of CLIP, it could also be used as a proxy source dataset. It would be interesting to ablate the effect of ImageNet versus LAION400M as proxy source dataset.
3. The authors use means and variances of tokens to align the distribution. I am wondering would these be sufficient or more sophisticated statistics or losses are needed for better alignment. I would expect more insights on why these two was used.

**Questions:**

Please refer to the weaknesses section for details.

---

> ### Author Rebuttal · Authors · 2023-08-09
>
> ## **Using higher-order statistics for alignment**
>
> Higher-order statistical alignment is a classical approach in test-time domain adaptation. However, since we align distributions computed using a small set of samples, we use the relatively simpler (mean and variance) measures for our alignment strategy. We evaluate the choice of higher-order statistical measures for the distribution alignment using the Central Moment Discrepancy measure [1]. However, we note that the increment is minimal thus using the mean and standard deviation for alignment.
>
> ### Comparison of PromptAlign vs Higher order statistics for Alignment
>
> | Method|Flowers|DTD|Pets|Cars|UCF|Caltech|Food|SUN   |Aircraft|Eurosat| Avg |
> |-----|-----|---|---|---|---|-----|----|---|-----|-----|----|
> | PromptAlign | 72.39  | 47.24  | 90.76  | 68.5  | 69.47  | 94.01  | 86.65  | 67.54  | 24.80  | 47.88  |66.92 |
> | w/ CMD  | 72.36  | 47.24  | 90.74  | 68.3  | 69.48  | 94.04 |  86.65 | 67.85  |  24.75 | 48.50 |66.99 |
>
>
> ## **Utilizing more samples for distribution alignment**
>
> We experiment with the use of more samples for distribution alignment. For each test example, we use a bag of samples (N=5) obtained from the same class (while the class label is kept unknown) to estimate the test sample statistics. We present the results in the table below which shows that test sample statistics with more samples can be better approximated, improving the performance through the alignment objective. However, in the actual setting of test-time adaptation, only a single sample is available.
>
> ### Bag of Samples Comparison
>
> | Method|Flowers|DTD|Pets|Cars|UCF|Caltech|Food|SUN   |Aircraft|Eurosat| Avg |
> |-----|-----|---|---|---|---|-----|----|---|-----|-----|----|
> | PromptAlign | 72.39  | 47.24  | 90.76  | 68.5  | 69.47  | 94.01  | 86.65  | 67.54  | 24.8  | 47.88  |66.92 |
> | PromptAlign*  | 74.49  | 51.52  | 91.60  | 74.00  | 74.81  | 96.12 |  87.67 | 68.83  |  25.68 | 50.86 |69.56 |
>
> PromptAlign* is fed with a bag of samples (N=5) along with the original test sample at the test time.
>
> ## **Experiments with LAION400M dataset statistics**
>
> As suggested, we ablate on the performance of the alignment strategy using LAION400M to compute the source data statistics. Given the time constraint, we use a subset of LAION400M with 2.5 million images (2x of ImageNet). The similarity of LAION400M to the CLIP training dataset yields a better statistical measure of the distributions, thus improving the distribution alignment objective and thus the performance. Given the large scale of the LAION400M dataset and since CLIP achieves excellent zero-shot performance for ImageNet, we found ImageNet as a feasible choice as the proxy source dataset which is consistent with previous work [2]. We will include these results in the paper.
>
> ### LAION vs. Imagenet statistics for distribution alignment loss.
>
> | Method|Flowers|DTD|Pets|Cars|UCF|Caltech|Food|SUN   |Aircraft|Eurosat| Avg |
> |-----|-----|---|---|---|---|-----|----|---|-----|-----|----|
> | ImageNet | 72.39  | 47.24  | 90.76  | 68.5  | 69.47  | 94.01  | 86.65  | 67.54  | 24.8  | 47.88  |66.92 |
> | LAION  | 72.38  | 47.97  | 90.76  | 68.11  | 70.2  | 94.32 |  87.26 | 68.02  |  24.78 | 47.88 |67.17 |
>
> [1]: Central Moment Discrepancy (CMD) for Domain-Invariant Representation Learning
>
> [2]: Bahng, Hyojin, et al. "Exploring visual prompts for adapting large-scale models." arXiv preprint arXiv:2203.17274 (2022).

---

> > ### Comment · Reviewer_wdgw · 2023-08-21
> >
> > Thanks for the responses, my main concerns are addressed and thus will increase my rating from 'Borderline accept' to 'Weak Accept'.

---

### Author Rebuttal · Authors · 2023-08-09

We sincerely thank the reviewers (wdgw, H4M3, ftF2, JL6L) for their detailed and positive feedback. The idea of improving test-time adaptation is interesting (ftF2), holds practical value (H4M3), and the proposed distribution alignment in V-L models at test time is novel (JL6L) and technically sound (wdgw). The proposed method is extensively evaluated on an array of benchmarks (wdgw, H4M3, JL6L) and achieves good results on various settings (H4M3, ftF2, JL6L). The paper is well-written and easy to read (wdgw, H4M3, ftF2, JL6L).

We validate the distribution alignment strategy further in terms of the statistical measures used for alignment, and the effect of improved statistics using a bag of samples to estimate test-sample statistics. Table 1 provides the results of using a bag of samples for our test-time adaptation strategy. We also analyze the effect of the proxy source dataset compared to the use of LAION400M and validate the choice of ImageNet as a valid proxy source dataset. The extended cross-dataset evaluation including LAION statistics for alignment is reported in Table 2. In addition, we extend our method to the base-to-novel benchmark setting and show consistent improvement in this setting as well, in Table 3. We also report the computational cost in terms of time and FLOPS for comparison with the baseline Test-Time Prompt Tuning [1] in Table 4. Our code and pre-trained models will be publicly released. All suggested changes will be reflected in the paper.

### Table 1. Bag of Samples Comparison

| Method|Flowers|DTD|Pets|Cars|UCF|Caltech|Food|SUN   |Aircraft|Eurosat| Avg |
|-----|-----|---|---|---|---|-----|----|---|-----|-----|----|
| PrompAlign | 72.39  | 47.24  | 90.76  | 68.5  | 69.47  | 94.01  | 86.65  | 67.54  | 24.8  | 47.88  |66.92 |
| PromptAlign*  | 74.49  | 51.52  | 91.60  | 74.00  | 74.81  | 96.12 |  87.67 | 68.83  |  25.68 | 50.86 |69.56 |

### Table 2. Cross-dataset benchmark evaluation.

| Method | Caltech | Pets | Cars | Flowers | Food | Aircraft | SUN | DTD  | Eurosat |UCF| Average |
|-----|-----|---|---|---|---|-----|----|---|-----|-----|----|
| CLIP |  93.35 | 88.25 | 65.48 | 67.44 | 83.65 | 23.67 | 62.59 | 44.27 | 42.01 | 65.13 | 63.58 |
| CLIP+TPT | 94.16 | 87.79 | 66.87 | 68.98 | 84.67 | 24.78 | 65.50 | 47.75 | 42.44 | 68.04 | 65.10 |
| CoOp | 93.70 | 89.14 | 64.51 | 68.71 | 85.30 | 18.47 | 64.15 | 41.92 | 46.39 | 66.55 | 63.88 |
| CoCoOp | 93.79 | 90.46 | 64.90 | 70.85 | 83.97 | 22.29 | 66.89 | 45.45 | 39.23 | 68.44 | 64.63 |
| ProDA | 86.70 | 88.20 | 60.10 | 77.50 | 80.80 | 22.20 | - | 50.90 | 58.50 | - | 65.62 |
| MaPLe | 93.53 | 90.49 | 65.57 | 72.23 | 86.20 | 24.74 | 67.01 | 46.49 | 48.06 | 68.69 | 66.30 |
| MaPLe+TPT | 93.59 | 90.72 | 66.50 | 72.37 | 86.64 | 24.70 | 67.54 | 45.87 | 47.84 | 69.19  |66.92 |
| PromptAlign | 94.01 | 90.76 | 68.50 | 72.39 | 86.65 | 24.80 |67.54 | 47.24 | 47.86 | 69.47 | 66.92 |
| PromptAlign*  | 94.32 | 90.76 | 68.11 | 72.38 | 87.26 |24.78 | 68.02 | 47.97 | 47.88 | 70.20 | 67.17 |

PromptAlign* refers to LAION400M being used as the proxy source dataset

### Table 3. Base to novel benchmark evaluation

|   Dataset   | Split | ProDA | MaPLe |MaPLe+TPT| PromptAlign |
|------|------|------|-----|-----|------|
| ImageNet | Base  | 75.40 | 76.67 |	77.73 |	78.26  |
|  | Novel  | 70.23 | 70.54 |	72.24 |	72.59  |
| Flowers | Base  | 97.70 | 95.93 |	96.24 |	96.61 |
|  | Novel  | 68.68 | 72.40 |	72.10 |	72.34  |
| DTD | Base  | 80.67| 80.30 |	82.20 |	82.60  |
|  | Novel  | 56.48 | 59.23 |	59.91 |	60.55 |
| Pets | Base  | 95.43 | 95.43 | 95.23 | 95.38  |
|  | Novel  | 97.83 | 97.80 |	97.37 |	97.56  |
| Cars | Base  | 74.70 | 72.90 | 74.00 |	75.02  |
|  | Novel  | 71.20 | 73.97 |	75.20 |	75.71  |
| UCF101 | Base  | 85.23 | 82.97 | 82.23 |	84.11  |
|  | Novel  | 71.97 | 78.57 | 77.34 |	79.30  |
| Caltech101 | Base  | 98.27 | 98.00 |	98.54 |	98.60  |
|  | Novel  | 93.23 | 94.27 | 94.29 |	94.50  |
| Food101 | Base  | 90.30 | 90.70 |	91.13 |	91.63  |
|  | Novel  | 88.57 | 92.07 |	92.03 |	92.68  |
| Sun397 | Base  | 78.67 | 80.80 |	81.15 |	81.57  |
|  | Novel  | 76.93 | 78.70 |	79.18 | 79.48  |
| Aircraft | Base  | 36.90 | 37.27 |	34.31 |	37.21  |
|  | Novel  | 34.13 | 35.53 |	35.81 |	37.27 |
| Eurosat | Base  | 83.90 | 93.63 |	91.02 |	94.10  |
|  | Novel  | 66.00 | 72.87 |	68.96 |	72.71  |
| Average | Base  | 81.56 | 82.24 |	82.16 |	83.19  (+0.95)|
|  | Novel  | 72.30 | 75.09 |	74.95 |	75.88 (+0.79)|

### Table 4. Comparison of FLOPS and inference time.

|   Method   | GFLOPS | Inference time (secs) |
|------|------|------|
| MaPLe+TPT | 1751  | 0.197 |
| PromptAlign | 1752  | 0.216 |

[1]: Shu, Manli, et al. "Test-time prompt tuning for zero-shot generalization in vision-language models." Advances in Neural Information Processing Systems 35 (2022): 14274-14289.

---

### Decision · Program_Chairs · 2023-09-21

**Decision:**

Accept (poster)

**Comment:**

This paper received slightly diverging reviews. Reviewer ftF2 expressed some concerns from different aspects whereas the rest 3 reviewers acknowledge the contribution of this work overall. Some of the concerns from ftF2 were addressed after rebuttal but the main concern on novelty (similarity to TPT and the reuse of existing components) remains. Upon reading through the paper and all the discussions, the AC feels that this work contains enough contributions to be accepted despite not being groundbreaking. 1) The authors have done a decent set of experiments, and the results look promising. The effectiveness of the proposed method is thus sufficiently validated. 2) Although this is not the first work on TPT, the AC considers TPT a general problem setting rather than a specific method. Method wise, this work seems a novel application of CMD/Deep Coral to TPT. In light of this consideration, the AC recommends acceptance.

The decision of this paper was communicated with the SAC, and was made based on the discussion.